# Sleep deprivation detected by voice analysis

**Etienne Thoret** [1,2,3]*, **Thomas Andrillon**[4,5,6], **Caroline Gauriau**[5,6], **Damien Léger**[5,6‡], **Daniel Pressnitzer** [1‡]

**1** Laboratoire des systèmes perceptifs, Département d'études cognitives, École normale supérieure, PSL University, CNRS, Paris, France, **2** Aix-Marseille University, CNRS, Institut de Neurosciences de la Timone (INT) UMR7289, Perception Representation Image Sound Music (PRISM) UMR7061, Laboratoire d'Informatique et Systèmes (LIS) UMR7020, Marseille, France, **3** Institute of Language Communication and the Brain, Aix-Marseille University, Marseille, France, **4** Sorbonne Université, Institut du Cerveau - Paris Brain Institute - ICM, Mov'it team, Inserm, CNRS, Paris, France, **5** Université Paris Cité, VIFASOM, ERC 7330, Vigilance Fatigue Sommeil et santé publique, Paris, France, **6** APHP, Hôtel-Dieu, Centre du Sommeil et de la Vigilance, Paris, France

‡ These authors joint last first authorship on this work.
* etienne.thoret@univ-amu.fr

**Data Availability Statement:** The analyses and figures of the manuscript can be replicated with the scripts openly available at https://github.com/EtienneTho/privavox The Spectro-Temporal Modulations (STMs) model adapted from the NSL

## Abstract

Sleep deprivation has an ever-increasing impact on individuals and societies. Yet, to date, there is no quick and objective test for sleep deprivation. Here, we used automated acoustic analyses of the voice to detect sleep deprivation. Building on current machine-learning approaches, we focused on interpretability by introducing two novel ideas: the use of a fully generic auditory representation as input feature space, combined with an interpretation technique based on reverse correlation. The auditory representation consisted of a spectro-temporal modulation analysis derived from neurophysiology. The interpretation method aimed to reveal the regions of the auditory representation that supported the classifiers' decisions. Results showed that generic auditory features could be used to detect sleep deprivation successfully, with an accuracy comparable to state-of-the-art speech features. Furthermore, the interpretation revealed two distinct effects of sleep deprivation on the voice: changes in slow temporal modulations related to prosody and changes in spectral features related to voice quality. Importantly, the relative balance of the two effects varied widely across individuals, even though the amount of sleep deprivation was controlled, thus confirming the need to characterize sleep deprivation at the individual level. Moreover, while the prosody factor correlated with subjective sleepiness reports, the voice quality factor did not, consistent with the presence of both explicit and implicit consequences of sleep deprivation. Overall, the findings show that individual effects of sleep deprivation may be observed in vocal biomarkers. Future investigations correlating such markers with objective physiological measures of sleep deprivation could enable "sleep stethoscopes" for the cost-effective diagnosis of the individual effects of sleep deprivation.

## Author summary

Sleep deprivation has an ever-increasing impact on individuals and societies, from accidents to chronic conditions costing billions to health systems. Yet, to date, there is no

toolbox (33) is available at: https://github.com/EtienneTho/strf-like-model.

**Funding:** Author ET was supported by grants ANR-16-CONV-0002 (ILCB), ANR-11-LABX-0036 (BLRI) and the Excellence Initiative of Aix-Marseille University (A*MIDEX) (ET). Author DP was supported by grants ANR-22-CE28-0023-01, ANR-19-CE28-0019-01, and ANR-17-EURE-0017. Author TA was supported by a Human Frontier Science Program Long-Term Fellowship (LT000362/2018-L). The funders had no role in study design, data collection and analysis, decision to publish, or preparation of the manuscript.

**Competing interests:** The authors have declared that no competing interests exist.

quick and objective test for sleep deprivation. We show that sleep deprivation can be detected at the individual level with voice recordings. Importantly, we focused on interpretability, which allowed us to identify two independent effects of sleep deprivation on the voice: changes in prosody and changes in voice quality or timbre. The results also revealed a striking variability in individual reactions to the same deprivation, further confirming the need to consider the effects of sleep deprivation at the individual level. Vocal markers could be correlated to specific underlying physiological factors in future studies, outlining possible cost-effective and non-invasive "sleep stethoscopes".

## Introduction

In the last decade or so, insufficient sleep has become a prominent public health issue, with one third of the adult population sleeping less than six hours per night [1–3]. This chronic sleep debt is associated with an increased risk of chronic disease, such as obesity, type 2 diabetes, cardiovascular diseases, inflammation, addictions, accidents and cancer [4–8]. Sleep debt also increase the risk of developing multiple comorbidities [9]. Moreover, more than one worker out of five operates at night and suffers from a high level of sleep deprivation [10], which causes accidents in the workplace or when driving [11]. In the present study, we use automated acoustic analyses of the voice to detect sleep deprivation. The aim is not to improve the accuracy of current machine-learning approaches [12–14], but, rather, to build on them to introduce a new focus on interpretability. Ideally, our method should not only detect whether an individual is sleep deprived or not, but also help to formulate specific hypotheses as to the physiological consequences of sleep deprivation for a given individual at a given moment in time.

Currently, there are several techniques aiming to measure sleep deprivation and its associated physiological consequences. First, sleep deprivation may be simply assessed in terms of the loss of sleep time, as measured in hours. Remarkably, however, the impact of a given amount of sleep deprivation varies massively across individuals. In laboratory settings where the amount of deprivation could be precisely controlled, up to 90% of the variance in cognitive performance was related to individual traits and not to the actual time spent asleep [15,16]. Second, sleep deprivation may also be measured through subjective sleepiness, which participants can explicitly report using rating scales [17–19]. However, subjective sleepiness could be influenced by other factors than sleep deprivation, such as the time of the day, motivation, or stress. Besides, it is not clear whether reported subjective sleepiness captures the full physiological impact of sleep deprivation, given the variety of the potentially implicit processes involved [20]. Third, objective methods have been developed to measure tangible consequences of sleep deprivation. The multiple sleep latency test [21], the gold standard in clinical settings, uses electro-encephalography (EEG) to estimate sleep latency (*e.g.* the amount of time to go from wake to sleep) along five successive naps sampled every two hours during daytime. The psychomotor vigilance test [22], often used in research settings, tests for the ability to respond quickly to infrequent stimuli, with slower reaction times assumed to be markers of attentional lapses. More recently, new approaches have attempted to measure the concentration of key molecules in the urine, saliva or breath [23]. Although these objective methods are complementary to subjective reports, they are often costly, time consuming, or difficult to deploy outside of laboratories. So, whereas there are cheap and fast objective diagnosis tools for other causes of temporary cognitive impairment, such as alcohol or drug abuse, there is currently no

established means to estimate sleep deprivation effects, at the individual level, in real-life settings.

If sleep deprivation could be detected through voice recordings, this would fill this gap by providing a quick, non-invasive, and cost-effective objective measure. Indeed, because the voice is easy to record with off-the-shelf equipment, there is a growing interest in finding vocal biomarkers to diagnose a variety of medical conditions [24,25]. For sleep, the idea was first explored by Morris et al. [26]. Free speech was produced by sleep deprived participants and rated by the authors. A slowing down of speech and a "flatness of the voice" were noted after deprivation. These observations were extended by Harrison and Horne [27], who found that raters blind to the amount of deprivation of the speakers could detect effects on the intonation of speech after deprivation. More recently, an experiment using a larger database found that, indeed, raters could detect sleepy versus non sleepy voices with an accuracy above 90% [28]. So, it does seem that there are acoustic cues in the voice that reflect sleep deprivation and/or sleepiness.

Machine learning has been applied to automate the detection of sleep deprivation and/or sleepiness from the voice. In an early study [29], sleep deprivation was inferred with high accuracy from vocal recordings (86%) but it should be noted that the deprivation was extreme, consisting of 60 hours without sleep, with unknown applicability to the much more common situation of mild sleep deprivation. Two "computational paralinguistic challenges" have since been launched, with sub-challenges aimed at assessing sleepiness from vocal recordings [30,31]. We will not review all of the entries to these challenges here, as they are quite technical in nature. To summarize, all of them used a similar framework: i) selection of a set of acoustic features, such as pitch, spectral and cepstral coefficients, duration estimates, and functionals of those features; ii) dimensionality reduction of the feature set; iii) supervised learning of target classification using various machine learning techniques, such as support vector machines or neural networks. The best results varied depending on the challenge. Subjective sleepiness proved difficult to predict [28], but the binary categorization of sleepy versus non-sleepy voices could be achieved with high accuracy (over 80%) in the best performing classifiers [32].

The framework described above will be familiar -and effective- for many machine learning problems, but it has two major limitations from a neuroscientific perspective. First, the initial selection of features is based on a somewhat arbitrary choice. Often, the choice of features was guided by the "voice flatness" hypothesis [26,27]. However, other, perhaps more subtle acoustic markers of sleep deprivation or sleepiness may have been overlooked by human raters. Second, the acoustic features discovered by the classifiers are not necessarily interpretable and can be difficult to relate to plausible mediating mechanisms [14]. Interestingly, the best-performing system so far used a carefully hand-crafted small feature set inspired from auditory processing models, suggesting that "perceptual" features may be a promising route for sleepiness detection in the voice [32]. A more recent study has again attempted to focus on "simple" acoustic descriptors for one of the databases of the paralinguistic challenge, with the explicit aim to facilitate interpretation [33]. Accurate classification was possible with the simpler feature set of about 20 features, with a resulting accuracy of 76%.

Here, we aim to extend these findings in four different ways. First, we use our own vocal database, which has been collected in a controlled laboratory setting where the amount of sleep deprivation could be precisely controlled. Vocal recordings were obtained from reading out loud the same texts for all participants, in random order across participants. This is important to avoid biases confounding sleep deprivation with *e.g.* participant identity, which is easily picked up by classifiers [28,34]. Second, we use a fully generic acoustic feature set, spectro-temporal modulations (STMs), computed with a neurophysiologically-inspired model of auditory processing [35]. The STM representation has been successfully applied to various machine-

learning problems such as musical instruments classification [36], timbre perception [36,37], or speech detection and enhancement [38]. Third, we apply our own technique to interpret the cues discovered by the classifier [39]. This technique, similar in spirit to the reverse correlation method used in neuroscience and psychophysics, identifies the parts of the input representation that have the most weight in the classifiers' decisions. The main outcome of the analysis consists in determining the parts of auditory feature space impacted by sleep deprivation. Fourth, by fitting classifiers to individual participants, we aim to uncover plausible physiological factors underlying the large and as of yet unexplained variability observed in the responses to mild sleep deprivation in normal healthy adults.

## Results

Twenty-two healthy women between 30–50 years of age (42.7 ± 6.8) were sleep deprived during a controlled laboratory protocol. An all-female experimental group was chosen because the current experiment took place in parallel with a dermatology study [40], but also because such a choice was expected to homogenize the vocal pitch range across participants. After a first "Control night" spent in the laboratory, participants were restricted to no more than 3 hours of sleep per night during two subsequent "Restriction nights", also monitored in the laboratory. Such a sleep restriction is both more ecological than total sleep deprivation and better controlled than observational paradigms. Vocal recordings were obtained throughout the protocol, during six reading sessions. The first three reading sessions occurred at different times of the day right after the control night (no sleep deprivation). The last three reading sessions occurred at the same times of the day but right after the second restriction night (see Methods for further details). All participants read, for about 10 minutes, chapters of the same French classic book: "Le Comte de Monte Christo" by Alexandre Dumas. The order of the excerpts was randomized across sessions for each participant to avoid a confound with deprivation. In total, our database thus consisted of 22 healthy participants producing about half an hour of vocal recordings each (M = 31min, SD = 5min) evenly split between before and after two nights of mild sleep deprivation. Each recording session was further split into 15-s frames. The main classification task was to decide whether a frame was recorded before deprivation or after deprivation.

### Subjective sleepiness reports are highly variable

Sleepiness was self-reported by participants at four different times during the day (similar but distinct from the three times when voice recordings were obtained, see Methods) using the Stanford Sleepiness Scale (SSS) questionnaire [19]. Fig 1A shows the distributions of SSS ratings. On average, sleep deprivation had an effect on self-reported sleepiness: sleepiness was low right after the control night, but increased after the deprivation nights. This was confirmed by an ANOVA on the SSS, with factors Day (2 levels, before and after deprivation) and Time of Report (4 levels). Both factors had a significant effect, but with a much larger effect size for Day ($F(1,46) = 52.14$, $p<0.001$, $\eta_p^2 = 0.221$) compared to Time of report ($F(3,92) = 3.07$, $p = 0.029$, $\eta_p^2 = 0.048$). Moreover, there was no interaction between Day and Time of report ($F(3,92) = 0.59$, $p = 0.621$). Because of this lack of interaction, we now consider average SSS values for all Times of Reports in a Day, to focus on the effect of sleep deprivation.

Fig 1B illustrates the data aggregated in that way, with individual changes in sleepiness now identified across the control and sleep deprived day. A remarkable individual variability was obvious in reported sleepiness. Note that this was in spite of our precise control of the amount of sleep deprivation, which was equated across all participants. Even so, some participants showed little effect of sleep deprivation, with even cases of *decreases* in subjective sleepiness *after* deprivation. Such unexpected effects were observed for all baseline sleepiness, low or

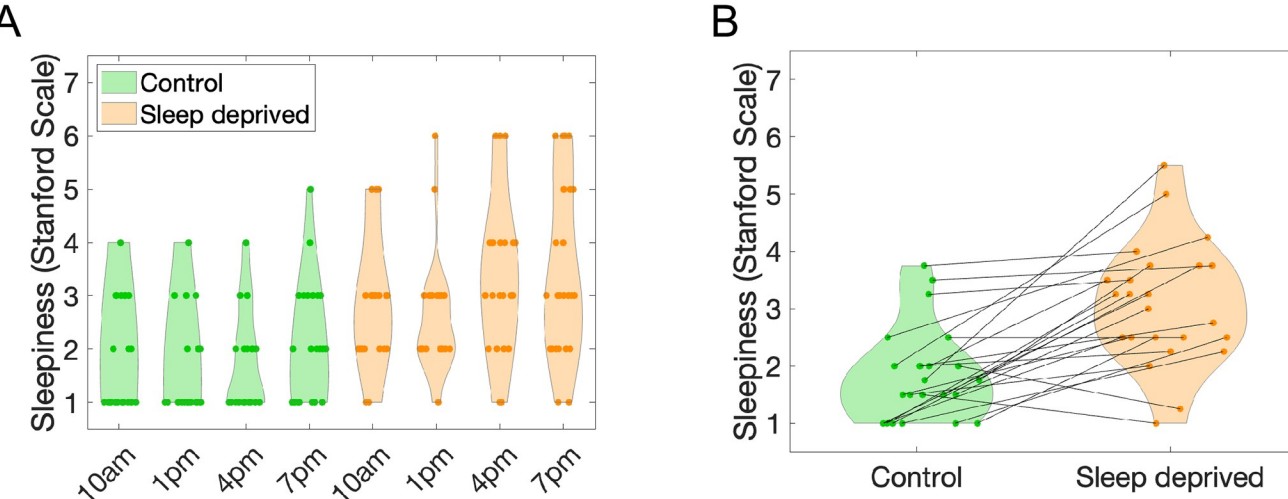

**Fig 1. A. Subjective sleepiness**. Sleepiness was evaluated by self-reports on the Stanford Scale before sleep deprivation (Control) and after two nights of mild sleep deprivation (Sleep deprived). The abscissa indicates the time of day when sleepiness reports were collected. **B. Average reported sleepiness before and after sleep restriction**. Lines connect data points for each participant.

high, as measured before deprivation. This striking variability is in fact consistent with previous observations involving objective measures of sleep deprivation [16]. It also further justifies that vocal biomarkers of sleep deprivation should be investigated at the individual level.

## Acoustic features of speech before and after sleep deprivation are broadly similar

To get a first qualitative overview of the effects of sleep deprivation on the acoustic features of speech, and in particular to test whether deprivation had any obvious average effect on the voice, we computed STM representations before and after deprivation.

Let us describe the STM representation in more details. At each moment in time, STMs contain the dimensions of *frequency*, *rate*, and *scale*. The *frequency* dimension, in Hz but on a logarithmic scale, reflects the spectral content of the sound. It is obtained by bandpass filtering the temporal waveform into several sub-bands, in a way intended to simulate peripheral auditory filtering. Rates and scales are then obtained using a bank of spectro-temporal modulation filters at the output of each peripheral channel. The *rate* dimension, in Hz, reflects the modulations in sound amplitude in the time domain. Slow modulations have low rates, whereas fast modulations have high rates. Positive rates indicate temporal modulations coupled with downward changes in frequency, whereas negative rates indicate temporal modulations coupled with upward changes in frequency. The *scale* dimension, in cycle per octave, reflects modulations in the spectral domain. Sounds with fine details in their spectral envelopes have high scale values, while sounds with relatively flat spectral envelopes have low scale values. For speech, the dominant rates are between 2 Hz and 8 Hz [41], while dominant scales, related to the harmonic structure of vowels, are around 2 cyc/oct [42]. The STM representation is similar to other widely used auditory front-ends such as the Modulation Power Spectrum (MPS) [43,44], the main difference being the logarithmic vs linear frequency scales used by the STM and MPS models, respectively.

The full STMs thus have four dimensions of time, frequency, rate, and scale. To have a look at the overall effect of sleep deprivation on acoustic features, we averaged the STMs along the

time dimension, separately before and after deprivation. Average STMs before (Fig 2A) and after (Fig 2B) deprivation were qualitatively similar. The rate-scale projections showed that, unsurprisingly, high energy in the STMs was focused in regions associated to speech [38]. The frequency-rate projection simply showed the average spectrum of our vocal recordings.

To further investigate the candidate acoustic differences caused by deprivation, we subtracted STMs before and after deprivation (Fig 2C for the population-level results, S1 Fig for individual-level results). At the population level, maximal differences in the rate-scale projection were less than 3%, while differences up to 11% were observed in the frequency-rate projection. At the subject level, differences in the rate-scale projection were around 24.68% on average (SD = 6), while differences up to 40.83% on average (SD = 12) were observed in the frequency-rate projection. Larger differences seem therefore observable at individual level but there is no obvious structure to the differences: they appear noisy and do not necessarily match the STM regions of high energy in speech (see S1 Fig).

For comparison with the state-of-the art of sleepiness detection from the voice [33], we also computed speech features using the openSMILE library [45]. The full 4368 speech features suggested in [33] were extracted (see Methods). Four of them are illustrated in Fig 2D, averaged before and after deprivation. These features were selected according to the "voice flatness" hypothesis. According to this hypothesis, it could be that sleep deprivation lowered the average pitch of the voice and reduced with its variation. It could also be that the quality of the voice, described with such adjectives as "creakiness" or "breathiness", could systematically change after deprivation. The closest openSMILE correlates of such perceptual descriptors are shown in Fig 2D. Visually, no obvious change was induced by sleep deprivation, with increases or decreases for all four features.

At this point, it is unclear whether the raw acoustic differences illustrated in Fig 2 are meaningful compared to the within- and across-participant variability. Also, the choice to illustrate 4 features out of 4368 is somewhat arbitrary. So, it remains to be tested whether the STM or openSMILE features have any predictive power to detect sleep deprivation. To address this point in a principled manner, we now turn to machine-learning, for the new STM representation and also for the openSMILE feature set.

## Detection of sleep deprivation from the voice is possible based on generic auditory features

A first question raised by the present study is whether fully generic auditory features can be used to detect sleep deprivation from the voice. To address this question, the STM representation was used as the input feature space for a standard machine-learning pipeline [13,14,36]. Recordings were converted to 15-s long frames, to be classified as "before sleep deprivation" or "after sleep deprivation" on the basis of their STM representation. The dataset was first transformed into train and test splits. We then reduced the high dimensionality of the feature space by means of a principal component analysis (PCA) on the training set (see Methods). The PCA spaces were finally fed to a support vector machine classifier (SVM) with a Gaussian kernel (radial basis function). We opted for an SVM and not a deep-learning architecture mainly because of the relatively modest size of our dataset, but also because SVMs have been shown to outperform more complex classifiers in similar tasks [14]. The performance of the SVM was evaluated with Balanced Accuracy (BAcc, see Methods).

At the population level, two cross-validation strategies were used. First, a Leave-One-Subject-Out (LOSO) strategy, in which one subject was left out of the training set and constituted the test set. The procedure was repeated for each participant. This procedure is the most stringent test of generalization of prediction for unknown participants. However, in small datasets

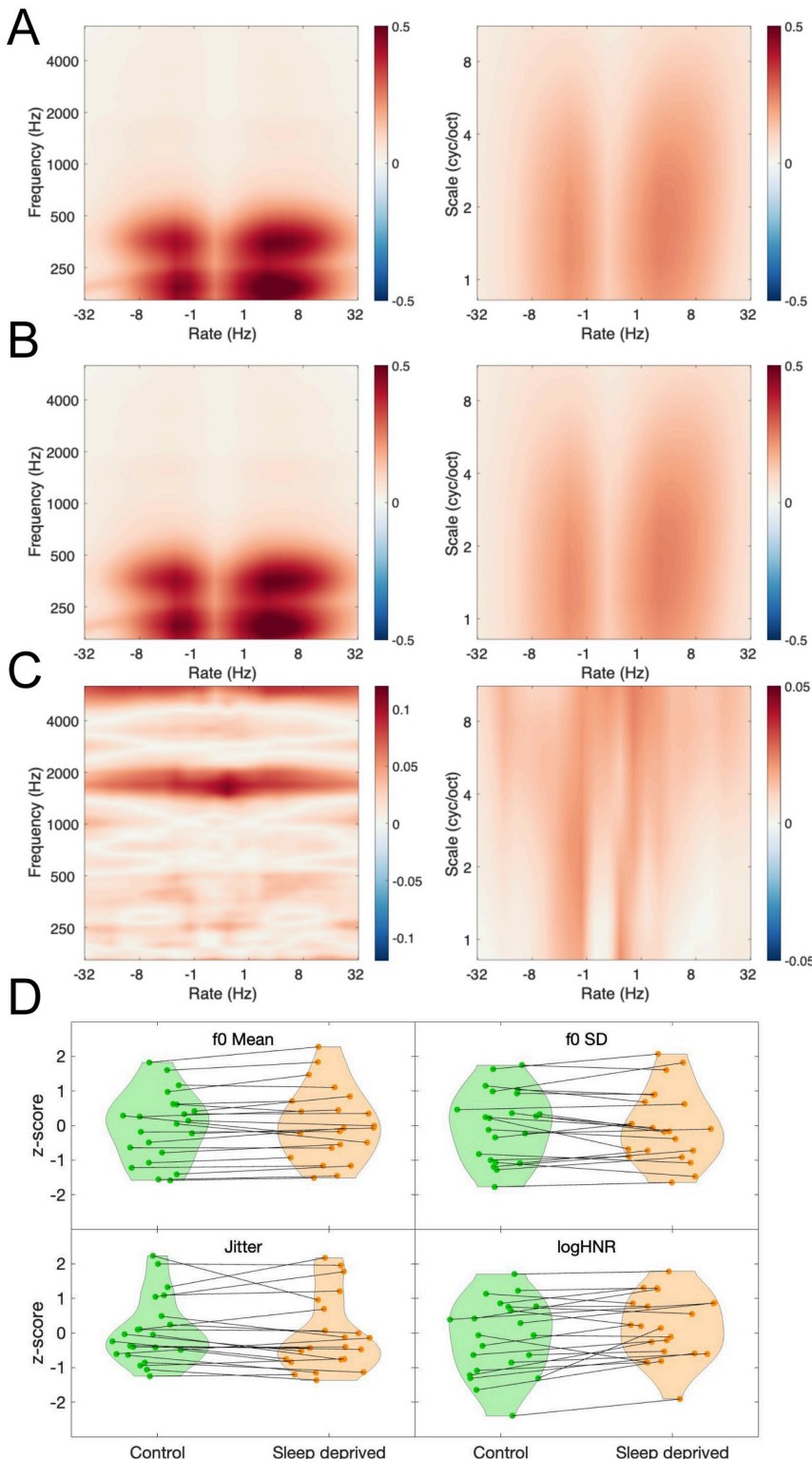

**Fig 2. Acoustic analyses. A**. Spectro-Temporal Modulations before sleep deprivation. Projections on the rate-scale and rate-frequency planes are shown. Arbitrary model units. **B**. As in A., but after sleep deprivation. **C**. Acoustic difference before and after sleep deprivation, shown as 2 * abs(B-A) / (A+B). Units of percent. **D**. Speech features before (green) and after (orange) sleep deprivation. Displayed are four openSMILE features related to average pitch (mean of the fundamental frequency $f_0$), pitch variation (standard deviation of $f_0$), voice creakiness (jitter) and voice breathiness (logarithm of the Harmonic to Noise Ratio). Lines connect data points for each participant.

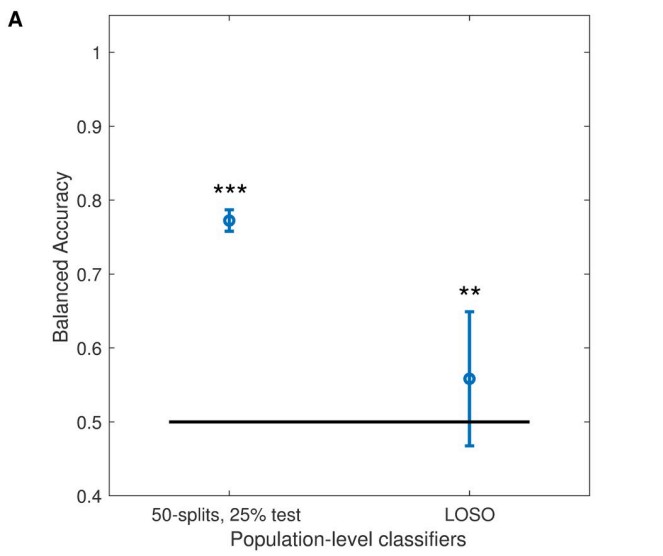
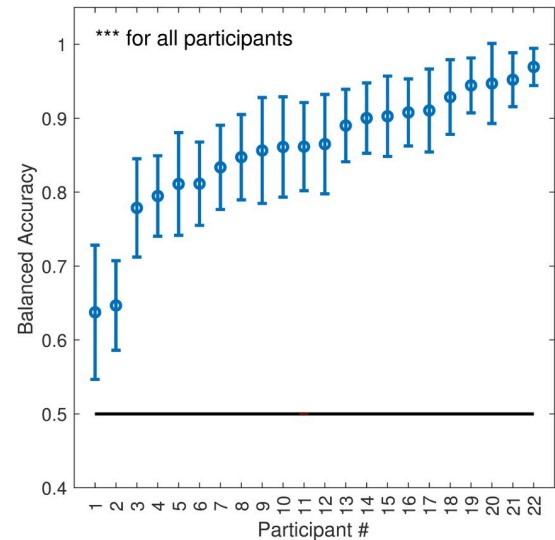

**Fig 3. Machine learning classification results with STM input features. A**. Balanced Accuracies for the population-level classifier using the generic STM representation as input feature space. Two cross-validation procedures are reported (see text). Error bars show standard deviations. Stars indicate the significance level of *t*-tests against chance level (** < .01; *** < .001). **B**. Balanced Accuracies for the classifiers tuned to individual participants, obtained with the 50-splits, 25% test cross-validation procedure. Participants are ranked according to classification accuracy.

with a large amount of individual variability, it has been argued that LOSO may be inappropriate [46]. This is likely the case for our dataset, with 22 participants and a large expected variability for the effects of sleep deprivation. Thus, we also report cross-validation using a 50-times repeated splitting of 25% of the data (50-splits, 25% test) randomly selected among the whole pool of participants, as suggested in [46]. At the participant level, the LOSO strategy does not make sense, so only the 50-splits, 25% test validation was used.

Classification performance is shown in Fig 3. At the population level, the classifier was able to detect sleep deprivation significantly above chance (50-splits, 20% test: BAcc, M = .77, SD = .01, t-test against .5: $t(49) = 145.27$ p < 0.001; LOSO: BAcc, M = .56, SD = .09, *t*-test against .5: $t(21) = 3.01$, p = .006). This seems on par with the state of the art obtained with different speech databases [32,33]. Interestingly, and as expected from the sizeable individual variability observed in the SSS reports, the same machine-learning pipeline was more accurate when applied at the individual level (BAcc, M = .86, SD = .09). Noticeably, for half of the participants, the classifiers' accuracies displayed BAccs above .9, outperforming the state of the art and matching human performance on a similar task [28]. For two participants, the classifiers' accuracies were relatively poor. Participant #1 displayed a decrease in sleepiness after deprivation (-0.75 for the sleepiness ratings averaged after and before deprivation), and was the only participant to exhibit such a trend in the group for which vocal recordings were available (another participant exhibited such a decrease in Fig 1B, but could not be included in the vocal analysis because of missing recordings). This may have contributed to the poor accuracy of the classifier. Participant #2 did exhibit an increase in sleepiness after deprivation (+0.75), so there are no obvious reasons for the classifier's poor performance in this case.

Overall, classification accuracies show that there is enough information in the fully generic STM representation of vocal recordings to detect mild sleep deprivation in otherwise normal and healthy participants. The classification performance at the population level is poor using a LOSO cross-validation procedure, and it is also possible that the population classifiers

benefited from a high input dimensionality to learn several individual recordings from different features sets, so the generalizability of our approach across speakers is not warranted. However, performance is generally excellent at the individual level, strengthening the idea that individual variability is key when considering vocal correlates of sleep deprivation.

### Individual classifiers detect voice changes related to sleep deprivation

Individual classifiers were excellent at discriminating recordings before and after deprivation, but there is one critical question that needs be addressed before any further investigation: were the classifiers sensitive to changes in the voice related to the experimental manipulation, sleep deprivation, or were they rather only picking up random acoustic variations that are bound to exist across different reading sessions?

Some sources of random acoustic variation can be ruled out by our experimental protocol. All participants were recorded with the same microphone before and after deprivation, in the same room, at the same location in the room. This minimizes confounds such as microphone frequency response or room reverberation.

However, there remains the possibility that participants simply talked in different ways when reading the same text more than once, for reasons unrelated to sleep deprivation. To test for this possibility, we took advantage of the availability of multiple reading sessions per participant: three reading sessions before deprivation and three reading sessions after deprivation, recorded at different times of the day. The before and after sessions were pooled together for the main analysis of Fig 3. Here, we trained and cross-validated individual classifiers using the same pipeline as for the main analysis, but this time to discriminate between all possible pairs of different reading sessions for a given participant. We then averaged the average accuracy for all sessions "Within" the same state (discriminate two different sessions recorded both before or both after deprivation) or "Across" states (discriminate one session recorded before deprivation and one session recorded after deprivation). If our experimental manipulation had no effect, the accuracies "Within" and "Across" should be identical.

This was not the case (Fig 4, two-tailed comparison $t(21) = 5.1$, $p = 5e-5$). Note that the analysis likely underestimates the effect of the experimental manipulation, because a lot of "Across" accuracies were near ceiling (hence the use of Rationalized Arcsine Units for display and statistical analysis purposes, see [47]). Also, there are interesting, non-random causes of variability within a state: the circadian rhythm and the homeostatic pressure for sleep that are both associated with fatigue, which could explain the relatively high performance already in the "Within" case. In any case, the results show that the classifiers picked up changes in voice due to sleep deprivation.

### Using standard speech features does not improve sleep deprivation detection accuracy

Even if the STM representation successfully supported sleep detection at the individual level, it could be that it missed important speech features such as "pitch" or "pitch variation", which are at the core of the "voice flatness" hypothesis and are part of most automatic sleepiness detection pipelines.

To investigate this possibility, we used the full openSMILE feature set (4368 features) as input feature space. We then applied the same classification pipeline as for the STM representation, consisting of dimensionality reduction followed by a Gaussian kernel SVM. This resulted in a pipeline matching the state-of-the art for sleepiness [33] while allowing comparison between the two input spaces.

Results are displayed in Fig 5. At the population level, the openSMILE classifier accuracy was similar to the STM classifier (50-splits, 20% test: BAcc, M = .70 SD = .01, $t$-test against .5: $t$

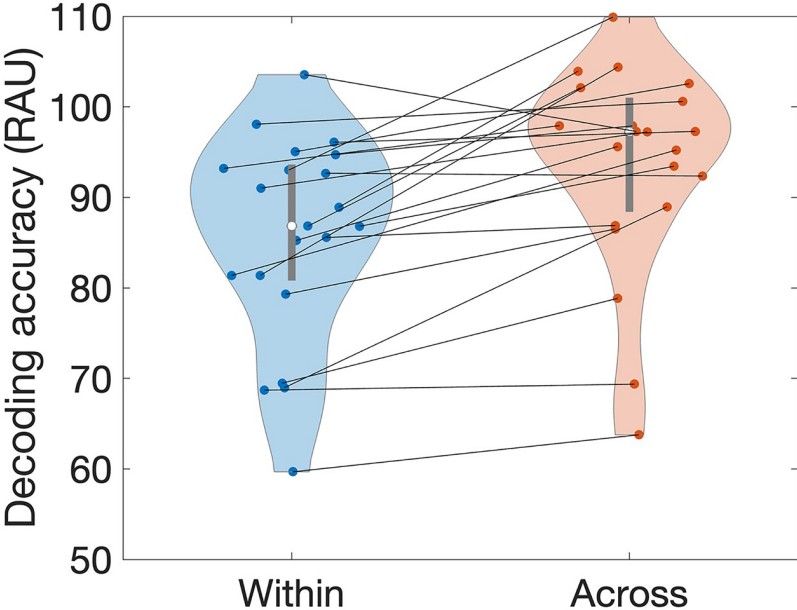

**Fig 4. Classification accuracy for voice variability unrelated versus related to sleep deprivation.** Balanced Accuracies for discriminating reading sessions recorded both before or after sleep deprivation ("Within") or for discriminating one reading sessions recorded before and one reading session recorded after sleep deprivation ("Across"). As 15 out of 22 classifiers produced accuracies above 0.9 for the Across comparison, BAccs were converted to rationalized arcsine units (RAU). Points represent individual participants. The median and interquartile intervals are also shown.

(49) = 91.6, p < 0.01; LOSO: BAcc, M = .56, SD = .09, *t*-test against .5: *t*(21) = 2.98, p = .007). At the individual level, the accuracies of the openSMILE classifiers were on average poorer than those observed with the STM classifiers (BAcc, M = 0.67, SD = 0.9). The correlation between classification performance using STM or openSMILE feature was low (r(20) = .34, p = .11). Interestingly, however, participants #1 and #2 for whom poor classification performance was observed using the STM input feature space also displayed poor classification using the openSMILE input feature space.

These results show that, for our voice database at least, using standard speech features decreased the accuracy of sleep deprivation detection. The relevant information to detect sleep deprivation from the voice was thus better expressed in the STM representation, with the added benefit, from our perspective, that generic auditory features should be easier to interpret. We now focus on the STM representation to interpret the features used for classification.

## Interpreting classifiers to identify vocal biomarkers of sleep deprivation

To gain insight about the nature of the acoustic features distinguishing speech before and after sleep deprivation, we probed the trained STM classifiers using a recent interpretation technique based on reverse correlation [39]. Briefly, the technique consists in randomly perturbing the input to the trained classifier, over thousands of trials, and then averaging all of the noisy representations leading to correct classification. This aims to identify the portion of the input that participates the most to the classifier's performance. The input representation was perturbed using additive noise in the PCA-reduced feature space [48]. Averaging all masks leading to a correct classification decision revealed, in our case, the discriminative features of a voice after deprivation compared to before deprivation (for details, see Methods and 39).

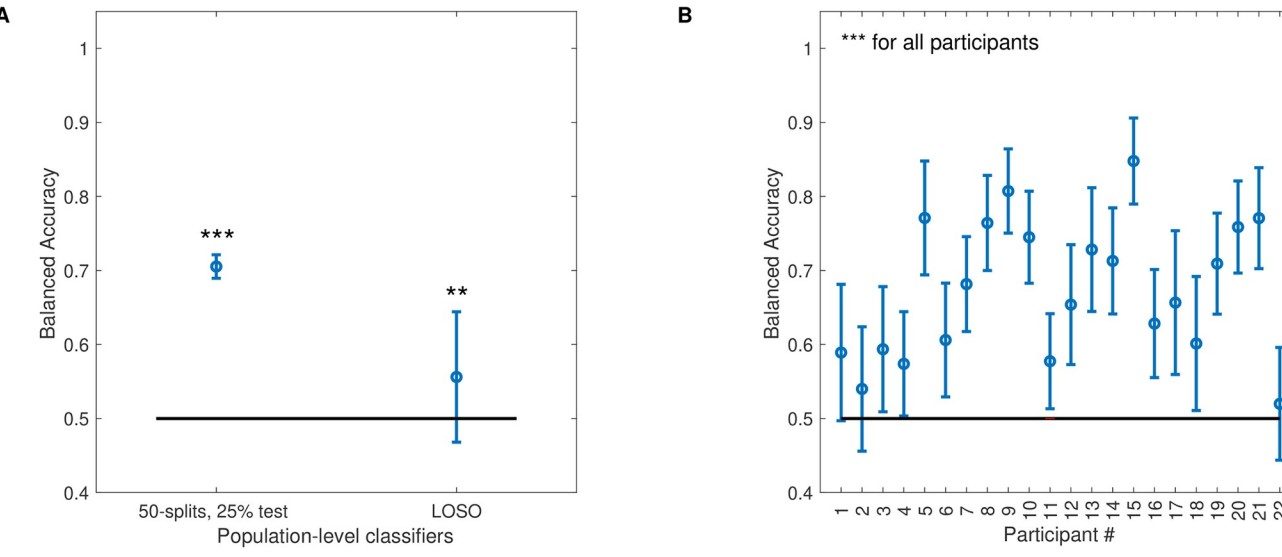

**Fig 5. Machine learning classification results with openSMILE input features.** Format as in Fig 3. In particular, for **B**., participants' labels (#) are identical to Fig 3.

As a preliminary step, we evaluated the consistency of the interpretation masks. Because of our cross-validation technique, 50 classifiers were fitted either for the whole dataset for the population-level classifier or for each participant's classifier. To check internal consistency, we computed the pairwise Pearson's correlation coefficients between all 50 interpretation maps. At the population-level, this "consistency correlation" was low albeit significantly above chance (r(22527): M = .20, SD = .34; all but 28 over 1225 pairwise correlations were significant, p < .05) which is consistent with the large variability suspected across listeners. At the participant-level, however, consistency correlations were very high (r(22527): M = .91, SD = .06, min = .73; all but 3 over 26950 pairwise correlations were significant, p < .05). Furthermore, because individual classifiers varied in accuracy, we could check whether the consistency of the interpretation improved with accuracy. As expected, the correlation between BAccs and consistency correlation was strong (r(20) = .71, p = .0003). These consistency results confirm that caution should be applied when considering population-level interpretations, but that individual results are robust and can be interpreted.

Fig 6 shows the interpretation maps for the population-level classifier. Maps should be read as follows: *red* areas correspond to STM features where the *presence* of energy is associated with sleep deprivation for the classifier, whereas *blue* areas represent STM features where the *absence* of energy is associated to sleep deprivation for the classifier. For the population-level map, the rate-scale projection resembles the raw difference before and after deprivation, although less noisy, whereas the frequency-rate projection does not match such raw acoustic differences (compare with Fig 2C). As these population-level interpretations are not robust, we simply show them for illustrative purposes and refrain from further description of their features.

Fig 7A shows all individual classifiers on the rate-scale projection, ordered along increasing accuracy (BAcc) of the corresponding classifier. We chose to interpret in priority the rate-scale projections, as is done in most speech applications [38]. The frequency-rate projections are provided as S3 Fig. The main feature of the results is the striking diversity of the individual maps, which is not related to classifier accuracy in any obvious manner. For some participants,

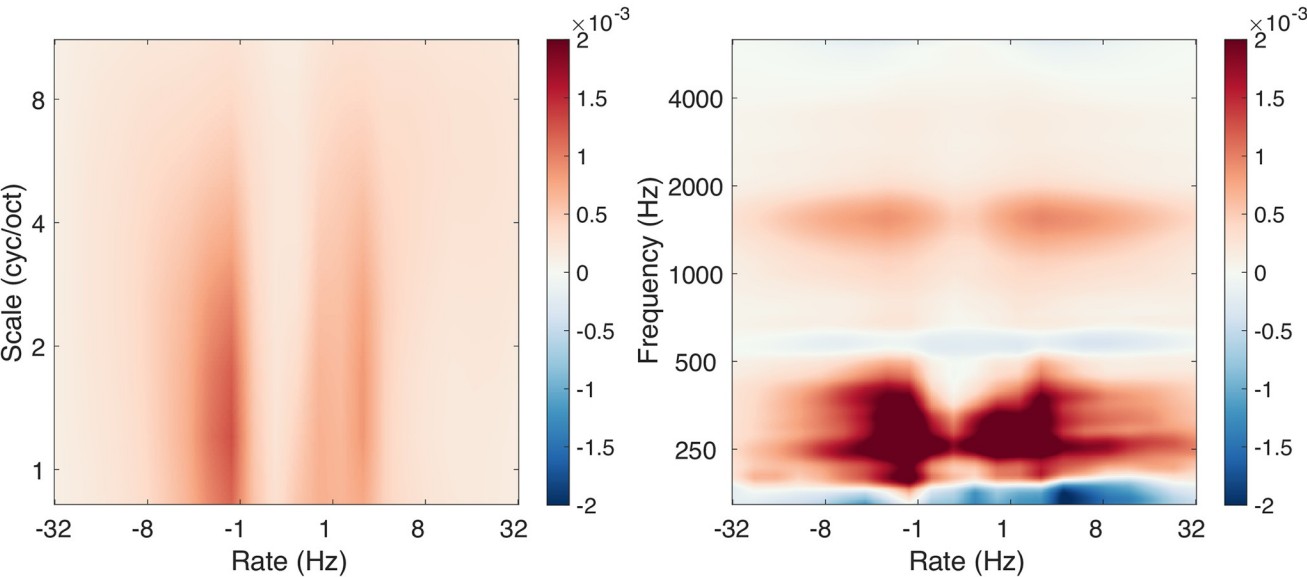

**Fig 6. Interpretation of the population-level classifier.** Discriminative features (see main text) are shown in the input STM space, for the rate-scale and frequency-scale projections. Red areas indicate features positively associated to sleep deprivation by the classifier. Blue areas correspond to features negatively associated to sleep deprivation by the classifier. Color bar indicate the averaged value of the reverse correlation mask. Values are low because of the relative low consistency of the interpretation masks for this population-level classifier.

sleep deprivation was detected through a reduction in energy over a range of STM features (blue color), consistent with a "flattening" of speech modulations. But the opposite was also observed for other participants (red color). Moreover, the details of the discriminative features also varied across participants. As shown before, these details are robust and warrant interpretation.

To get a better understanding of this variability across individual maps, we performed a PCA on the maps themselves, which we will term interpretation-PCA for clarity. A first interpretation-PCA dimension explained 35.9% of the variance, while a second dimension explained 24.2% of the variance. There was a drop for all other dimensions (N = 3) which explained less that 13% of the variance, see S4 Fig. Participants ordered on the first two interpretation-PCA dimensions are shown in Fig 6B. We computed the variance of all STM features along each interpretation-PCA dimension, to visualize the features that distinguished the interpretation maps along these main axes of variation. Results are shown in Fig 7C and 7D. The features defining the first interpretation-PCA dimension were clustered between rates of about 2 Hz to 8 Hz, which is exactly the amplitude modulation range corresponding to speech prosody and syllabic rate [41]. This shows that the amplitude modulation characteristics of speech was affected by sleep deprivation. Importantly, depending on the individual, the classifiers used the presence *or* absence of energy around these rates to detect sleep deprivation. This shows that while some participants spoke in a "flattened" voice after deprivation, consistent with classic hypotheses [26,33], others instead spoke in a more "animated" voice after deprivation. The features defining the second interpretation-PCA dimension clustered at very low rates and covered a broad scale range, peaking at about 2 cyc/oct. This corresponds to long-term spectral characteristics of speech and vowel sounds. In speech, such voice-quality or timbre features are determined by the precise shape of the various resonators inside the vocal tract, such as the throat and nasal cavities: by filtering the sound produced by the vocal folds, resonators impose formants that impact the timbre of vowels and other speech sounds.

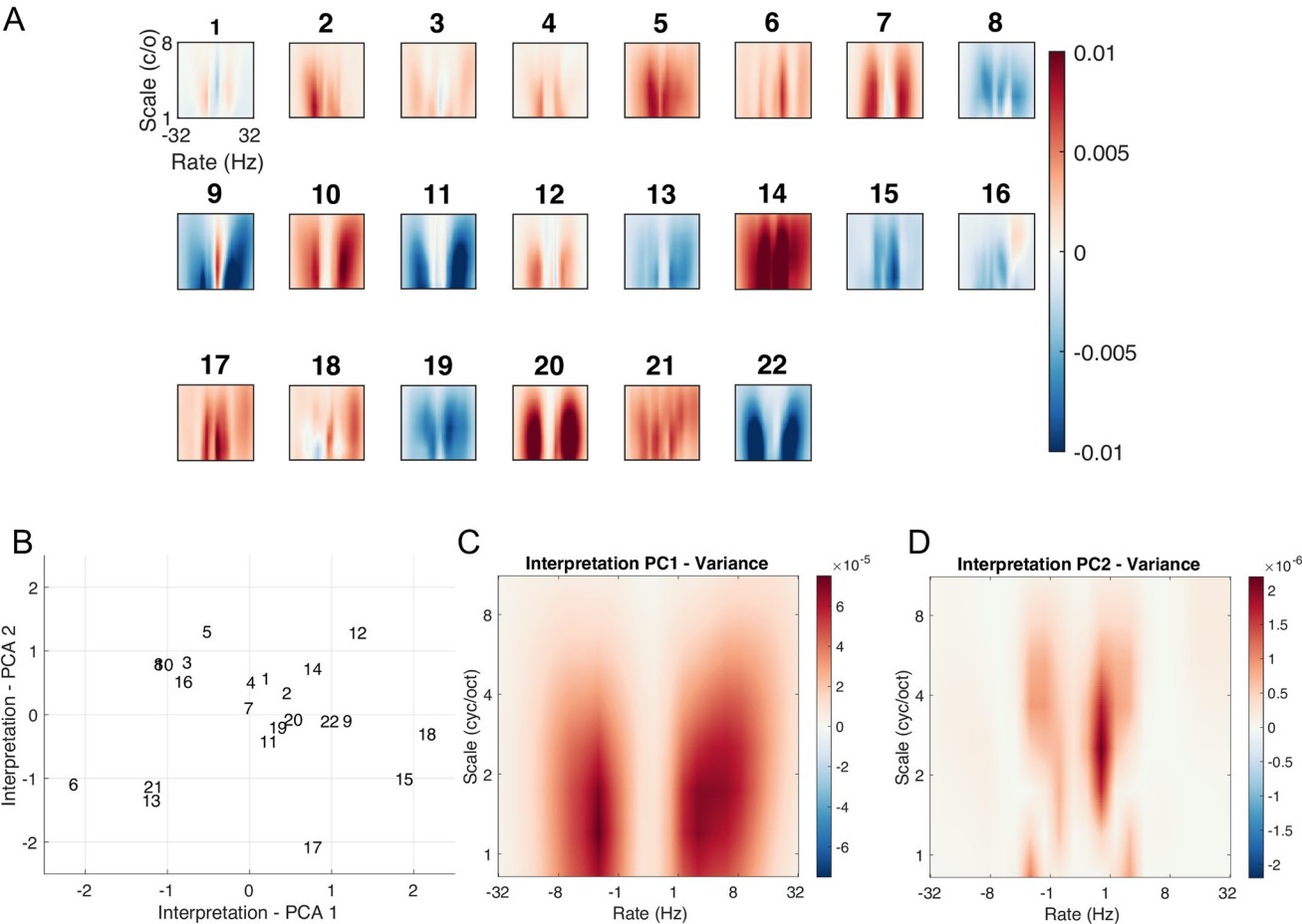

**Fig 7. Interpretation of the participant-level classifiers. A**. As for Fig 6, but for individual participants identified by their participant #. **B**. Projection of participants # in the interpretation-PCA space of all participant's masks (see text for details). **C. and D**. Variance of the idealized masks along the first two dimensions of the interpretation-PCA. Idealized masks are obtained by first sampling the PCA latent space between -2 and 2 for the two first dimensions with 30 values and then inverting the latent space into the input feature space by using the inverse transform of the PCA. Red areas show the discriminative features that vary the most along each interpretation-PCA dimension. Units: variance in the feature space.

## Correlation with subjective sleepiness reports

All participants were subjected to the exact same amount of sleep deprivation. Nevertheless, their subjective sleepiness reports varied widely (Fig 1). We investigated whether the variability in subjective sleepiness reports could be accounted for by characteristics of the individual machine-learning classifiers.

First, we simply correlated the individual classifier's accuracies to the individual SSS reports after sleep deprivation. If subjective sleepiness was a full measure of the impact of sleep deprivation, we would expect a high correlation between the classifier's accuracy and SSS reports. Results are shown in Fig 8A. There was no significant correlation between BAccs and SSS reports ($r(20) = .32$, $p = .14$, $BF_{10} = .47$), suggesting that subjective sleepiness did not express all of the objective effects of sleep deprivation, at least as captured by our voice classifiers.

Next, we investigated whether the classifier's interpretation maps could account for the SSS reports variability. In particular, we reasoned that the prosodic and rhythmic changes captured by the first interpretation-PCA dimension could be due to cognitive factors, inducing flattened

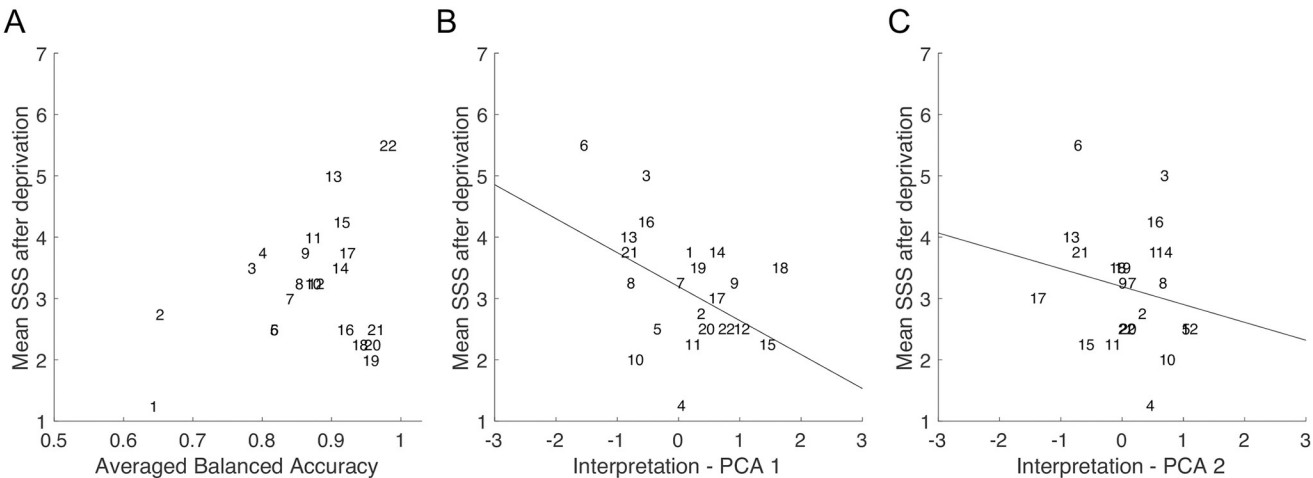

**Fig 8. Relation between subjective sleepiness and voice classifiers. A**. Subjective sleepiness is plotted as a function of balanced accuracy of each participant-level classifier. **B**. Subjective sleepiness is plotted as a function of the coordinate of each participant-level classifier on the first dimension of the interpretation-PCA space. **C**. As in B., but for the second dimension of the interpretation-PCA space.

or animated speech. Such factors could be explicit to participants–if only by self-monitoring their own speech rate and intonation. In contrast, the voice quality cues captured by the second interpretation-PCA dimension could be more subtle and remain implicit. Results are shown in Fig 8B and 8C. Consistent with our hypothesis, we observed a moderate but significant correlation between the location of participants on the first interpretation dimension and sleepiness reports ($r(20) = -.44$, $p = .03$, $BF_{10} = 1.34$). In contrast, the location of participants on the second interpretation dimension did not show any significant correlation with sleepiness reports ($r(20) = .19$, $p = .38$, $BF_{10} = .23$).

Finally, to assess the full information contained in the interpretation maps, we fitted a linear model that used coordinates on both interpretation-PCA dimensions to predict SSS scores after deprivation (see Methods). Results showed that it was possible to predict sleepiness from interpretation maps ($R^2$: M = .29, SD = .18) significantly above chance (two-sample t-test to 0: $p < .00001$). This results further suggests that the classifiers detected voice changes related to sleep deprivation and not random variation across reading sessions.

## Discussion

### Summary of findings

We ran a sleep deprivation protocol with normal and healthy participants, collecting subjective reports of sleepiness plus vocal recordings before and after deprivation. After two nights of mild sleep deprivation, subjective sleepiness increased on average, although with striking individual differences—including some participants even reporting decreases in subjective sleepiness after deprivation. Nevertheless, sleep deprivation could be detected accurately by means of machine-learning analysis of vocal recordings. Classification was most accurate at the individual level, with 85% balanced accuracy on average. Importantly, such a classification was based on a fully generic auditory representation. This allowed us to interpret the discriminative features discovered by classifiers to detect sleep deprivation. Two broad classes of features were revealed: changes in temporal modulations within the rhythmic range characteristic of speech sentences, and changes in spectral modulations within the timbre range of speech

sounds. Furthermore, the interpretation maps could account for some of the variability in subjective sleepiness reports, which were correlated to the changes in temporal modulations ("flattened" or "animated" voice).

## Candidate mechanisms underlying the vocal biomarkers of sleep deprivation

Our data-driven analysis revealed that classification was based on two classes of auditory features: temporal modulations in the 2 Hz to 8 Hz range, and spectral modulations around 2 cyc/oct. We now speculatively relate these vocal features to two classes of well-established neurophysiological effects of sleep deprivation.

The temporal modulation features associated to sleep deprivation were in a range which has been robustly found as characteristic of speech across a variety of languages, to the extent that they have been described as "universal rhythmic properties of human speech" [41]. Such a universal rhythm is imposed by the biomechanical constraints of the vocal apparatus and by the neurodynamics of its control and perception systems. The changes in speech rhythms observed after sleep deprivation could thus result from a temporary impairment of the cognitive control of the speech production process. Sleep deprivation impacts cognitive function [20], presumably through changes in glucose consumption in frontal and motor brain regions [49,50]. Accordingly, previous studies showed lower activity in the dorsolateral prefrontal cortex and in the intraparietal sulcus in cognitive tasks requiring attention, with large inter-individual variability [51]. A reduced connectivity was also observed within the default mode network, the dorsal attention network, and the auditory, visual and motor network following sleep deprivation [50,52,53]. Finally, extended wakefulness has been associated with an increase in the intrusion of sleep-like patterns of brain activity in wakefulness [54,55]. All these results suggest that sleep deprivation is akin to a minor cognitive frontal dysfunction, and may thus plausibly affect the fluency of vocal production. Interestingly, compensatory responses were also observed in cognitive tasks, which may explain why some of our participants responded to deprivation with less speech modulation, consistent with the classic "flattened voice" hypothesis [26,27], whereas others unexpectedly responded with speech over-modulation and instead produced an "animated voice" after deprivation.

The spectral modulation changes detected by our classifiers were consistent with changes in the timbre of speech sounds, and in particular vowel sounds [35,38,42]. Such sounds acquire their distinctive spectral envelopes by means of the resonances of the vocal tract, including the throat and nasal cavities. Inflammation of the throat and nose could be responsible for these changes in timbre. Sleep deprivation is known to trigger an immune response leading to inflammation. A cortisol increment can be observed after a single night of sleep deprivation [8,56,57], so is plausible in our protocol that included two nights of mild sleep deprivation. In terms of mechanisms, sleep restriction and deprivation disturb the normal secretion of hormones like cortisol or testosterone, and is associated with increased rates of interleukin-6 and CRP as assessed on salivary samples in normal subjects. This inflammatory response could be linked to an elevated blood pressure following sleep deprivation [58] and could affect the vocal tract and impact the spectral envelope of speech. It should be noted that other variables, such as changes in hydration or food intake due to deprivation, might also impact characteristics of the vocal apparatus and induce timbre changes instead or in addition to putative inflammation. Such additional variables were not controlled in our protocol.

## Limitations of the study

There are both technical and conceptual limitations to the present study. We chose to use a controlled protocol to precisely equate sleep deprivation in our participants, but this came at the expense of a relatively small dataset compared to the online databases used by machine-learning challenges [30,31]. Our protocol prevented biases in the database, such as associating the identity of speakers with the amount of sleep deprivation [28], but also limited our choice of possible machine-learning techniques to perform the classification. We thus used an SVM classifier, and not potentially more powerful deep-learning architectures. We note however that in the studies that compared SVMs with other classifier types, SVM performed best, including in state-of-the-art studies [14,32,33]. In any case, the interpretation method we used could be applied to any kind of classifier [39], including more complex ones.

All participants were female, mainly for practical reasons. Sleep deprivation might affect females and males differently, in particular with respect to inflammation, although the evidence is still mixed [59]. The generalizability of our findings to males thus remains to be tested experimentally. In addition, the modest performance observed for population-level classifiers limits the generalization of our approach to unknown speakers, which would be desirable for practical use cases involving pre-trained classifiers. However, this also confirms the interest to apply interpretation techniques at the individual level, to capture the variability that seems inherent to the effects of speech deprivation.

The feature set we used was a generic auditory representation, which is a major difference with previous machine-learning oriented studies. On the one hand, some studies were fully data-driven and selected the best-performing features from thousands of speech descriptors. The resulting features were often difficult to interpret. On the other hand, there were also studies using a small set of features, but these features were carefully hand-crafted and potentially lacked genericity. Our approach represents a trade-off between these two ideas: we applied a data-driven approach to select a small subset of features, but because these features were from a generic representation, they remained interpretable. A clear limitation is that we did not include features related to pitch or functionals of pitch such as contour features, which have been repeatedly shown to be useful for sleepiness detection [14,32,60]. However, average estimates of pitch and pitch variation (Fig 2D) suggested that there were no obvious effects on these features in our database. Furthermore, our classification pipeline applied to standard speech features performed worse than using the STM representation. We believe that these omissions were compensated by the richness of the STM representation. Pitch and pitch functionals will in fact be indirectly reflected in the STMs, which analyses sounds over a broad range of temporal scales simultaneously.

The possible physiological mechanisms that we put forward as a mediation between sleep deprivation and vocal features have to be considered as fully speculative for now. We did not collect the objective measures required to confirm or infirm these interpretations. The cognitive factor could be assessed with objective behavioral measures, such as the psychomotor vigilance test [22], or with brain imaging data [49,50]. The inflammatory factor could be assessed by biological analyses of *e.g.* cortisol in the saliva [8,57]. Because we have not gathered such measurements, we can only argue that both minor cognitive dysfunction and inflammation effects are likely for our participants as a group. In any case, the present study is the first one to suggest that such factors may be measured at the individual level from voice biomarkers, and it raises the possibility for future investigations to confirm or reject this hypothesis by actually correlating vocal features with more invasive objective markers.

The classification task we investigated consisted only in detecting whether a vocal recording was performed before or after sleep deprivation. We did not attempt to decode the effect of

more subtle factors on the voice, such as the time of the day, which would reflect the interactions between circadian rhythms and sleep deprivation. These interactions have been shown in a recent study [61], albeit using a more severe deprivation protocol (60 hours without sleep). Unfortunately, our experimental design does not provide the statistical power to examine within-day variations before sleep deprivation (half the dataset) or the interaction between within-day variations and deprivation (second-order effect). In the same study, a regression approach was implemented to provide predictions beyond binary classification. Interestingly, this approach was successful only for predicting objective measures, such as sleep latency, but failed for subjective reports. This is consistent with the claim that subjective scales incompletely characterize the full effects of sleep deprivation and can usefully be complemented by objective measures such as voice analysis. As we did not collect objective measures of sleepiness beyond the voice, we did not attempt a regression analysis.

Finally, on a conceptual level, we wish to raise a basic but inescapable limitation of any study of sleep deprivation. Sleep deprivation may be defined, as we did, by the amount of sleep available to each individual. However, as has been repeatedly pointed out and again observed here, there is a remarkable diversity of responses to the same amount of sleep deprivation. Thus, it should not be expected that any one measure will capture all of the effects of sleep deprivation. Subjective reports may capture explicit feelings of fatigue, but be blind to implicit effects [61]. With objective measures, which are by necessity indirect, there is an issue with interpreting negative outcomes. In our case for instance, how to interpret a relatively poor accuracy for a sleep deprivation classifier, such as was observed for two participants? It cannot be decided whether this poor accuracy showed that sleep deprivation had no effect on these participants, or that sleep deprivation had effects that were not expressed in the voice, or that the classifiers simply failed for technical reasons. Measuring multiple markers of sleep deprivation, including the novel ones we suggest, and incorporate them into a holistic model of the neurophysiological effects of sleep deprivation seems to be a promising way forward.

## Perspectives

Keeping these limitations in mind, the demonstration of vocal biomarkers for sleep deprivation could have major clinical implications. Subjective sleepiness reports do not capture the whole effect of a lack of sleep [61]. Moreover, such reports rely on the honest cooperation of participants, which is not a given if self-reports of excessive sleepiness can have negative work-related or financial consequences for the individual. Objective correlates of sleepiness exist [21,22], but vocal biomarkers would represent a considerably cheaper and faster alternative, requiring no specialized equipment and increasing their practicality for real-life clinical assessment. Crucially, our technique also goes beyond the simple binary detection of sleep deprivation: thanks to the application of interpretability techniques [39], we suggest that different neurophysiological processes related to sleep deprivation may be untangled through the voice alone. Such measures could in turn be used to design interventions tailored to each individual and situation, if the effects of sleep deprivation needed to be temporarily alleviated for instance. More generally, there is a growing realization that interpretability is key to future clinical applications of artificial intelligence, as both patients and clinicians would likely want to understand the reason for a diagnostic [62]. For application to real-life settings, it is particularly interesting to identify features that do not correlate with subjective sleepiness, as one of the biggest dangers of sleep loss is the partial agnosia for one's own sleepiness.

To finish, it is useful to point out that the methodological pipeline we introduced here is fully generic, as the audio features representation used is itself generic and the interpretation method can be applied to any classifier. Therefore, the present study could pave the way for

future investigations of vocal biomarkers over the broad range of fundamental or clinical applications that are currently only starting to be considered [24,25].

## Materials and Methods

### Ethics statement

The study was conducted according to French regulations on human research including agreements from the Hotel-Dieu Hospital Ethics Committee (CPP Ile de France 1—N˚ 2017-sept.-13690), with signed consent from participants who received financial compensation. Our protocol was conducted in accordance with the 2016 version of the Declaration of Helsinki and the ICH guidelines for Good Clinical Practice.

### Experimental design

A group of twenty-four healthy women between 30–50 years old (42.7 ± 6.8) took part in the experiment. This study was part of a dermatological study and only Caucasian phototypes I-IV (Fitzpatrick classification) were recruited. Participants were non-smokers and did not report a history of substance abuse. They had a Body Mass Index (BMI) between 19 and 25, no sleep disorders or chronic disease, no daytime vigilance issues (Epworth Sleepiness Scale $\leq$ 10), and were not under any medical treatment (exclusion criteria).

Before the experiment, participants wore an actigraph for 7 days and were instructed to maintain a regular sleep-wake behavior with their usual 7–8 h of sleep (i.e., in bed from 23:00–01:00 until 07:00–09:00). The compliance with these recommendations was verified through the actigraphic recordings (MW8, CamTech; UK) that were inspected by the research team at the participant's arrival the morning before the first night of sleep restriction (day 1). No sleep episodes were detected outside of the scheduled experimental time in bed (see [40] for details). The protocol lasted for 3 days (day 1: before sleep restriction; day 2: during sleep restriction; day 3: after sleep restriction), which included 2 nights of sleep deprivation (at the end of day 1 and 2). For days 1 and 2, participants were instructed to restrict their sleep time to 3h (i.e., in bed from 03:00 to 06:00) and to follow their usual routine outside the laboratory. After the second sleep-restricted night (day 3), the participants went to the laboratory on the morning and their actigraphy recordings were immediately analysed to ensure their compliance with the imposed sleep-wake hours. During day 1 (after habitual sleep and before sleep restriction: baseline condition) and day 3 of each session, the participants remained in the sleep laboratory from 09:00 to 19:00 under continuous supervision. In order to help the participants stay awake, from the moment they left the laboratory at the end of day 1 until their return to the laboratory at the beginning of day 3 at 09:00, two investigators exchanged text messages with the participants at random times during the entire period outside of the laboratory. Text messages were sent throughout the night (except during the period where participants were instructed to sleep, that is between 3 and 6 a.m.). Participants had to respond right after receiving these messages. In case of an absence of response, participants were immediately called on their personal phone. For lunch in the laboratory (days 1 and 3), participants received controlled meals consisting of a maximum of 2,500 calories/day with a balanced proportion of nutrients (protein, fat, and carbohydrates).

### Voice recordings

During day 1 (before sleep deprivation) and day 3 (after), at three different times during the day (9am, 3pm, 5 pm), participants were seated and instructed to read 10 minutes of different chapters of the same French classic book: "Le Comte de Monte Christo" (Alexandre Dumas,

1844). Their voice was recorded with a portable recorder (Zoom H1/MB, stereo-recording). Then, during one minute, participants produced free speech, but these recordings were not used in the present analyses. Two participants had to be discarded at this stage, as technical issues prevented the completion of all recording sessions.

## Baseline speech feature set

We computed basic speech features using the openSMILE library [45], in the configuration recommended for the Interspeech 2011 challenge. This feature set has been used for in state-of-the-art studies detecting sleepiness from voice recordings [33]. It consists of 59 low-level descriptors, including 4 energy descriptors, 50 spectral descriptors, and 5 voice descriptors. These descriptors were then combined with 33 base functionals and 5 $f_0$ functionals, resulting in a total of 4,368 features.

## Spectro-Temporal Modulations (STM)

The sound files, initially sampled at 44.1 kHz, were down-sampled to 16 kHz. Spectro-Temporal Modulations (STMs) were computed with our own toolkit, available on the repository associated with the paper and which is directly adapted from the standard NSL Toolbox [35]. Sounds were processed through a bank of 128 constant-Q asymmetric bandpass filters equally spaced on a logarithmic frequency scale spanning 5.3 octaves, which resulted in an auditory spectrogram, a two-dimensional time-frequency array. The STM were then computed by applying a spectro-temporal modulation filterbank to the auditory spectrogram. We generally followed the procedure detailed in [36], with minor adaptations. A 2D Fourier transform was first applied to the spectrogram resulting in a two-dimensional array [44] whose dimensions were spectral modulation (scale) and temporal modulation (rate). Then, the STM representation was derived by filtering the MPS according to different rates and scales and then transforming back to the time-frequency domain. We chose the following scale (*s*) and rate (*r*) center values as 2D Gaussian filters to generate the STMs: $s$ = [0.71, 1.0, 1.41, 2.00, 2.83, 4.00, 5.66, 8.00] cyc/oct, $r$ = ±[.25, .5, 1, 2, 4, 5.70, 8, 11.3, 16, 22.6, 32] Hz. Such a range covers the relevant spectral and temporal modulations of speech sounds as already used in different studies [63]. The resulting representation thus corresponds to a 4D matrix with dimensions of time, frequency, scale, and rate.

## Classification pipeline

For all recordings, STMs were computed and used as the basis for the input feature space. STMs were computed with 22 rates x 8 scales x 128 frequencies per 40ms temporal windows. One data point for the classifier consisted of the average of 3841 successive temporal windows, yielding *frames* of 15 seconds. Thus, there were 22528 features for every 15-s long frame.

Standard machine-learning pipeline were then used [13,14,36] to predict whether a frame belonged to the sleep deprived class or not. First, the whole dataset was randomly separated into a training set and a testing set, either by randomly holding 25% of the data into the testing set or, only at population level, by holding-out the data from one participant to define the training and the testing in a Leave-One-Subject-Out (LOSO) cross-validation procedure. We then reduced this high dimensionality of the feature space by means of a principal component analysis (PCA). At the population level, we trained a PCA on the whole dataset and retained the 250 main dimensions, explaining 99% of the variance. We further checked that the exact choice of PCA dimensions did not affect our conclusions, about the performance but also about the interpretation of the classifiers (see S2 Fig). At the participant level, for each participant we trained a PCA on the data from all other participants, to reduce a possible

contamination of the reduced space by peculiarities of the considered participant. We next retained the 30 main dimensions of the PCA. The number of PCA dimensions in this case was chosen empirically, so that the reduced feature space still explained more than 90% of the variance and provided a dimensionality lower than the number of frames available for each participant (between 98 and 194 frames of 15 s each), to avoid overfitting. We checked that the exact choice of PCA dimensions did not affect our conclusions, in particular on the interpreted features that are consistent for PCA dimensions above 30.

The PCA spaces were then fed to a support vector machine classifier (SVM) with a gaussian kernel (radial basis function). The training set was used to fit the SVM through an hyperparameter grid-search, using a stratified 5-folds cross-validation. The fitted SVM was then evaluated on the testing set by computing Balanced Accuracy (BAcc, defined as the average of true positive rate, or sensitivity, with true negative rate, or specificity). For the randomly selected train/test split, we repeated the fitting procedure 50 times, generating 50 distinct train/test sets for both the population and individual levels (denoted 50-splits, 25% test). In the Leave-One-Subject-Out (LOSO) approach, we replicated the fitting procedure with 22 participants, each excluded once and designated as the testing set. In each instance, we computed the final balanced accuracies and then averaged them across either the 50 different train/test splits (in the case of 50-splits, 25% test) or across the 22 subjects (in the case of LOSO). Lastly, for each cross-validation procedure, we conducted a t-test against the threshold of 0.5 using the distributions of balanced accuracies. This allowed us to evaluate the classifier's capability to predict sleepiness from voice, assessing its performance compared to random chance. All the classification pipelines from PCA to RBF + SVM are implemented with the scikit-learn library [64].

## Interpretation of the classifiers

Each classifier fitted in the study was probed with a reverse correlation technique which provides an interpretation of which features are relevant to the classifier [65]. We refer to our method paper for a full description of the method [39]. Briefly, noisy inputs or "probes" were generated and the classifier's decisions were registered for each probe. The average of probes leading to a "before deprivation" was computed and then subtracted to the average of probes leading to an "after deprivation" decision, to obtain what we termed an "interpretation mask" for the classifier. We used the version of the method where probes are pseudo-random noises in the PCA-reduced input space [39], as this increases search efficiency. For each classifier, the number of probes was set to 100 times the number of available frames, which represented between 9800 and 20000 probes depending on the classifier. The resulting interpretation masks are composed of positive and negative values. Positive values correspond to features which are associated to sleep deprivation, while negative values correspond to features associated to the absence of sleep deprivation. Here we refer to the content of the mask as mask as the "discriminative information", which has also been termed "represented features" previously [39].

## Supporting information

**S1 Fig. Acoustic difference before and after sleep deprivation before and after sleep deprivation computed as 2\*abs(After-Before) / (After+Before)–for each subject ranked according to the classifier accuracies.** Units: Percent.
(TIF)

**S2 Fig. Performance of the classifiers as a function of the number of principal components, averaged across the 22 subjects.** (Right) Average pairwise correlations between the

interpretation masks for different number of Principal Components averaged across the 22 subjects. In order to evaluate for how many PCs the interpretations are stabilized, pairwise correlations between each masks, 1 one for each PC, has been done for each and averaged across the whole 22 subjects. We observe that the interpretation is stabilized around 20 PCs which. (Left) Error bars represent standard deviations.
(TIF)

**S3 Fig. Average masks in the frequency-rate projection for all subjects (ranked by balanced accuracy).** Red indicates the areas of the frequency rate projections that are used by classifier to predict a sleep deprived excerpt and conversely, blue indicates frequency-rate areas that characterize a non-sleep deprived voice.
(TIF)

**S4 Fig. Explained variance with respect to the number of principal components for the interpretation-PCA analysis.** Units: percent.
(TIF)

## Acknowledgments

We would like to thank Andrey Anikin for helpful comments that led to the analysis presented in Fig 4.

## Author Contributions

**Conceptualization:** Etienne Thoret, Thomas Andrillon, Damien Léger, Daniel Pressnitzer.

**Data curation:** Etienne Thoret, Thomas Andrillon, Daniel Pressnitzer.

**Formal analysis:** Etienne Thoret, Thomas Andrillon, Daniel Pressnitzer.

**Funding acquisition:** Damien Léger, Daniel Pressnitzer.

**Investigation:** Etienne Thoret, Thomas Andrillon, Caroline Gauriau, Damien Léger, Daniel Pressnitzer.

**Methodology:** Etienne Thoret, Thomas Andrillon, Caroline Gauriau, Damien Léger, Daniel Pressnitzer.

**Project administration:** Etienne Thoret, Damien Léger, Daniel Pressnitzer.

**Resources:** Etienne Thoret, Thomas Andrillon, Damien Léger.

**Software:** Etienne Thoret.

**Supervision:** Etienne Thoret, Thomas Andrillon, Damien Léger, Daniel Pressnitzer.

**Validation:** Etienne Thoret, Thomas Andrillon, Damien Léger, Daniel Pressnitzer.

**Visualization:** Etienne Thoret, Thomas Andrillon, Daniel Pressnitzer.

**Writing – original draft:** Etienne Thoret, Thomas Andrillon, Damien Léger, Daniel Pressnitzer.

**Writing – review & editing:** Etienne Thoret, Thomas Andrillon, Damien Léger, Daniel Pressnitzer.

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
