## [Decision Letter · Decision Letter 0]

5 Jul 2023

Dear Dr. Thoret,

Thank you very much for submitting your manuscript "Sleep deprivation measured by voice analysis" for consideration at PLOS Computational Biology.

As with all papers reviewed by the journal, your manuscript was reviewed by members of the editorial board and by several independent reviewers. In light of the reviews (below this email), we would like to invite the resubmission of a significantly-revised version that takes into account the reviewers' comments.

We cannot make any decision about publication until we have seen the revised manuscript and your response to the reviewers' comments. Your revised manuscript is also likely to be sent to reviewers for further evaluation.

Sincerely,

Frédéric E. Theunissen

Academic Editor

PLOS Computational Biology

Marieke van Vugt

Section Editor

PLOS Computational Biology

Reviewer's Responses to Questions

**Comments to the Authors:**

Reviewer #1: This article discusses machine learning as a useful tool for objectively evaluating sleep deprivation. Overall, the manuscript is extremely well written, with substantial motivation toward the use of voice recordings as a simple, non-invasive, non-costly method to assess sleep deprivation. I have a series of minor questions and grammatical changes to improve the significance and readability of the work:

1. Conclusions of an inflammatory response in the throat and nose is a huge jump from altered timbre. For instance, these results could also be the effect of reduced or increased amplitude of vocal tract movement, such as in lazy or mumbled speech vs. clear speech, or perhaps due to differences in hydration pre/post sleep deprivation. To be clear, inflammatory response is one possibility, but it is also possible that these results are due to effects from the third-variable problem.

2. Could a lack of significant correlation between subjective sleepiness reports and timbre cues be the result of the third-variable problem (e.g., hydration, food intake) rather than (or in addition to) the hypothesized implicit inflammatory effect?

3. Although I agree with the reasoning for choosing an all-female experimental group, I am curious: Is there reason to suggest that these results are generalizable across sex (why or why not)?

4. Is there any correlation with results and participant age, typical amount of sleep per night, or usual bedtime?

5. Why might the relatively poor accuracies resulted in 2/22 participants?

Grammatical Recommendations:

Line 60: increases (or can increase)

Line 150: as “of” yet

Line 294: consists of

Line 380: I recommend changing “vocal cords” to “vocal folds”

Reviewer #2: PCOMPBIOL-D-23-00407

Sleep deprivation measured by voice analysis

E. Thoret et al

Reviewer’s comments

This manuscript describes an experiment by which speech recordings from 22 healthy adult female talkers were collected before and after two nights of restricted sleep and then statistical learning methods were applied to the audio to discriminate between the two occasions. In terms of originality the manuscript describes a new corpus unlike any existing corpus, and exploits a method for exploring which acoustic features were used by the classifier – a method I had not seen applied to this area before. In terms of innovation the method uses a variant of a previously developed modulation spectrum for feature extraction, and well developed methods for feature selection and pattern classification. In terms of importance, the manuscript does not compare the new method with a baseline approach on the new corpus, so it is difficult to know whether the new methods are in improvement to existing approaches. In terms of insight, the manuscript speculates that there are two different mechanisms by which sleep restriction affects speech: a cognitive mechanism and a physiological mechanism – while this is almost certainly true, there is very little evidence in the experimental outcomes to be able to draw this conclusion. In terms of rigour the method does have some significant weaknesses, in particular there is no cross-validation of individual speakers (unless I have misunderstood the method), so that the same speaker can be present in both training set and test set; really the method needs to use a leave-one-speaker-out cross-validation to get an appropriate estimate of classification accuracy for an unknown speaker. In terms of evidence for conclusions, the paper bases its understanding of the effect of sleep restriction on voice on a post-hoc method of correlating acoustic features with classifier outputs – however the classifier did not perform particularly well at detecting sleep restriction, and as mentioned, was not tested with appropriate cross-validation.

Overall, the authors should think more carefully about what they want to show with this corpus with maybe a focus on what changes occur in the speech signal which correlate with sleep restriction and whether there is evidence for two separate processes.

Here are some suggestions for improvements to the study:

1. Introduce a leave-one-speaker-out cross validation to the method so as to establish a performance figure for speaker-independent sleep restriction detection.

2. Consider introducing a state-of-the-art baseline method to establish the difficulty of the task. For example, OpenSMILE features and an SVM with leave-one-out cross-validation.

3. Consider plotting some basic voice features as a function of sleep restriction, e.g. pitch height, pitch variation, speaking rate, breathiness, and creakiness.

4. Look at how speech recordings vary within the day, not just before and after sleep restriction. Typically it is fairly easy to distinguish morning speech from evening speech, at least in a speaker dependent system. Differences within one day may be larger than the differences before and after sleep restriction.

5. There is a directly relevant study by Baykaner et al [Baykaner KR, Huckvale M, Whiteley I, Andreeva S, Ryumin O, "Predicting Fatigue and Psychophysiological Test Performance from Speech for Safety-Critical Environments", Frontiers in Bioengineering and Biotechnology, 3, 2015] in which changes to speech over a period of sleep deprivation are tracked using support-vector-regression. This paper also uses modulation spectrum features. It would be useful to compare the findings of this study with results obtained on the new corpus. In particular looking at the problem as a regression problem rather than a classification problem would seem to open up more downstream applications. Indeed, the current title of the manuscript suggests that the current article “measures” sleep deprivation, whereas in fact it simply classifies speakers as being recorded before or after a period of restricted sleep.

**Have the authors made all data and (if applicable) computational code underlying the findings in their manuscript fully available?**

Reviewer #1: Yes

Reviewer #2: **No: **Code is available, but not original audio recordings, so the results in the paper cannot be verified

PLOS authors have the option to publish the peer review history of their article (what does this mean?). If published, this will include your full peer review and any attached files.

Reviewer #1: No

Reviewer #2: No
---

## [Decision Letter · Decision Letter 1]

3 Dec 2023

Dear Dr. Thoret,

Thank you very much for submitting your manuscript "Sleep deprivation detected by voice analysis" for consideration at PLOS Computational Biology.

As with all papers reviewed by the journal, your manuscript was reviewed by members of the editorial board and by several independent reviewers. In light of the reviews (below this email), we would like to invite the resubmission of a significantly-revised version that takes into account the reviewers' comments.

Dear Etienne,

I had to get a third reviewer to look at you manuscript because the first two declined to re-review. It has been particularly difficult to find reviewers for this paper.

As you will see this reviewer is implicitly asking about sample size effects. I look forward to reading your response.

Best,

Frederic.

We cannot make any decision about publication until we have seen the revised manuscript and your response to the reviewers' comments. Your revised manuscript is also likely to be sent to reviewers for further evaluation.

Sincerely,

Frédéric E. Theunissen

Academic Editor

PLOS Computational Biology

Marieke van Vugt

Section Editor

PLOS Computational Biology

Dear Etienne,

I had to get a third reviewer to look at you manuscript because the first two declined to re-review. It has been particularly difficult to find reviewers for this paper.

As you will see this reviewer is implicitly asking about sample size effects. I look forward to reading your response.

Best,

Frederic.

Reviewer's Responses to Questions

**Comments to the Authors:**

Reviewer #3: Thoret et al. present a study that applies machine learning to the problem of detecting sleep deprivation from voice samples. First of all, I would like to stress that I really enjoyed reading the study. The methods are brilliant, and the acoustic and data analysis are of excellent quality. Using spectrotemporal modulation features and then reverse-correlating the classifiers are both original and promising techniques to add to the quiver of voice scientists. While I am not so familiar with literature on sleep deprivation, the theoretical grounding of the study and literature review seem very good as well. Unfortunately, I do see one major problem, namely the central claim that the presented models detect sleep deprivation per se – see the next comment. Perhaps the authors can refute this concern, and then all is well; but if not, then the results are in serious doubt.

MAJOR

line 511 "some of our participants responded to deprivation with less speech modulation, consistent with the classic “flattened voice” hypothesis (26, 27), whereas others unexpectedly responded with speech over-modulation and instead produced an “animated voice” after deprivation" A person reading the same text twice will presumably not have exactly the same intonation, speech rate, manner of articulation, etc. because they may also be generally more animated, tired, anxious and so on at the second reading (and not merely more sleep-deprived). Thus, some variation between the two readings is virtually guaranteed, and I don't see how it is possible to determine whether acoustic changes are caused by sleep deprivation or something else if (1) the direction of these changes varies across individuals and (2) each individual is recorded only once (before + after sleep deprivation). Specifically, the implicit assumption that each 15 s frame in a recording is an independent datapoint is problematic, and the per-individual classifiers in Fig. 3B can be interpreted as detectors of a particular recording rather than of sleep-deprived state. To prove that you detect specifically sleep deprivation, I think you would need either to demonstrate robust classification with LOSO (which you don’t) or to record the same individual multiple times in normal/sleep deprived state (and thus prove that there are indeed subject-specific, but stable acoustic markers of sleep deprivation). It's actually interesting that the population-level classifier (Fig. 3A) is fairly good at ~70%, but it could be because there aren't that many recordings and a vast number of predictors, so the same SVM model can learn to recognize multiple recordings from different variables (look at how different the individual classifiers are in Fig. 6A); if this interpretation is correct, the population-level accuracy should steadily decline as you increase the number of participants included in the model (as suggested by the better performance of per-individual models in Fig. 3B).

MINOR

Abstract "two distinct effects of sleep deprivation on the voice: a change in prosody and a change in timbre" In speech science it is more usual to talk about "voice quality" rather than "timbre". The term "prosody" may also need to be defined explicitly as different authors include different acoustic characteristics it it.

Because Methods are placed at the end of the paper, the key methods really need to be at least briefly explained in the Introduction and/or the Results. Personally, I was initially puzzled by the spectrotemporal modulation space, assuming you were talking about MPS until I checked the Methods section, because the present explanation (lines 142-145 and 211-221) is really brief. I suggest adding a more informative description along the lines of the paragraph "Spectro-Temporal Modulations of musical sounds" in Thoret 2021. Another key piece of information is what consitutes a datapoint - apparently, a 15 s frame, so each subject had between 98 and 194 datapoint. This is only mentioned on line 720, but without this information the reader may assume that there was in fact just one datapoint per recording, and then it's a mystery how the classifiers could be trained for each participant separately.

line 490 "The individual classifiers using the STM input features learnt to detect sleep deprivation with a high accuracy, matching human performance" This would be more convincing if coupled with a perceptual experiment performed on these recordings (or shorter parts thereof)

line 731 "for each cross-validation procedure, we conducted a t-test against the threshold of 0.5 using the distributions of balanced accuracies." A more powerful test for both the 50-splits and LOSO would be a GLMM modeling binary classifications of each 15 s frame

Reviewed by Andrey Anikin

**Have the authors made all data and (if applicable) computational code underlying the findings in their manuscript fully available?**

Reviewer #3: Yes

PLOS authors have the option to publish the peer review history of their article (what does this mean?). If published, this will include your full peer review and any attached files.

Reviewer #3: **Yes: **Andrey Anikin
---

## [Editor Report · Decision Letter 2]

22 Jan 2024

Dear Dr. Thoret,

We are pleased to inform you that your manuscript 'Sleep deprivation detected by voice analysis' has been provisionally accepted for publication in PLOS Computational Biology.

Best regards,

Frédéric E. Theunissen

Academic Editor

PLOS Computational Biology

Marieke van Vugt

Section Editor

PLOS Computational Biology

Dear Etienne and Daniel

Thank you for your resubmission. I see that you have addressed the final set of comments. I am sorry this took me a bit of time since your resubmission. It was a busy time.

I look forward to seeing your paper in the journal.

Best wishes,

Frederic Theunissen

---

## [Editor Report · Acceptance letter]

31 Jan 2024

PCOMPBIOL-D-23-00407R2 

Sleep deprivation detected by voice analysis

Dear Dr Thoret,

I am pleased to inform you that your manuscript has been formally accepted for publication in PLOS Computational Biology. Your manuscript is now with our production department and you will be notified of the publication date in due course.

With kind regards,

Judit Kozma
